# Suboccipital Muscles, Forward Head Posture, and Cervicogenic Dizziness

**DOI:** 10.3390/medicina58121791

**Published:** 2022-12-05

**Authors:** Yun-Hee Sung

**Affiliations:** Department of Physical Therapy, College of Health Sciences, Kyungnam University, Changwon 51767, Republic of Korea; sungpt97@kyungnam.ac.kr; Tel.: +82-505-999-2173

**Keywords:** cervicogenic dizziness, suboccipital muscles, forward head posture

## Abstract

Dizziness or vertigo can be caused by dysfunction of the vestibular or non-vestibular systems. The diagnosis, treatment, and mechanism of dizziness or vertigo caused by vestibular dysfunction have been described in detail. However, dizziness by the non-vestibular system, especially cervicogenic dizziness, is not well known. This paper explained the cervicogenic dizziness caused by abnormal sensory input with references to several studies. Among head and neck muscles, suboccipital muscles act as stabilizers and controllers of the head. Structural and functional changes of the suboccipital muscles can induce dizziness. Especially, myodural bridges and activation of trigger point stimulated by abnormal head posture may be associated with cervicogenic dizziness.

## 1. Introduction

Sensory balance and orientation are dependent on the integration of symmetrical inputs from the vestibular organs, visual and auditory cues, and proprioception in the nervous system [1]. Asymmetry in the afferent input or any dysfunction in these sensory organs can result in dizziness and imbalance [2,3]. Dizziness is one of the most common symptoms of patients visiting a medical center. The one-year prevalence of dizziness was 8.4% in the US adult population [4]. Patients with dizziness usually have only one diagnosis, but some have more than one diagnosis (approximately 3.7%). The most common diagnoses are benign paroxysmal positional vertigo, vestibular neuritis, and Meniere’s disease [5].

However, vestibular lesions are not the only causes of dizziness, which can be categorized as vestibular (central or peripheral) and non-vestibular. Common reasons for non-vestibular dizziness include visual disturbances, metabolic dysregulation, cardiovascular autonomic dysregulation, side effects of medications, and central nervous system, endocrine, and cervical lesions [6,7]. Among these, cervicogenic dizziness (CGD) is caused by trauma, inflammation, degeneration, or mechanical dysfunction of the cervical spine [6,8,9,10,11], and is characterized by unsteadiness, neck pain, stiffness, headache, dysphagia, nausea, visual disturbances, ear fullness, tinnitus, temporomandibular joint pain, and other psychological problems [7]. Many researchers divide the causes of cervicogenic dizziness into three major categories: irritation of sympathetic plexus, abnormal somatosensory input, and vertebral artery insufficiency [3,10]. Of these, many clinicians have no doubt about the link between dizziness and vertebral artery insufficiency [3,10,12]. Patients with the vertebrobasilar system compression complain that their most common symptoms are dizziness, vertigo, headache, vomit, diplopia, ataxia, imbalance, and weakness in both sides of the body [12,13]. The clinical manifestations vary from syncope to posterior circulation stroke, but vertebrobasilar symptoms or syncope with head-turning, known as rotational vertebrobasilar insufficiency or Bow Hunter’s Syndrome, are rare [3,14,15,16,17]. Assessment tools (vertebrobasilar insufficiency test, transcranial doppler ultrasound) for diagnosing dizziness caused by vertebral artery insufficiency are also well known [18,19].

Postural sway can be induced by poor head posture [20]. Accumulated abnormal stimulations by poor head posture are transmitted to the cervical structures, and dysfunction of the related structures may cause dizziness [21]. Specially, changes in the upper cervical spine may be more correlated with dizziness than those in the lower cervical spine [7]. A significant proportion of total cervical flexion, extension, or rotation occurs in the upper cervical spine (atlanto-occipital and atlanto-axial joints) [8,13,22]. An amount of 50% of all cervical proprioceptors are present in the joint capsules of C1–C3 [23]. Mechanoreceptors on the joint capsules in the cervical spine also play an important role in proprioception [24,25]. These mechanoreceptors are abundant in γ-muscle spindles located in the deepest layers of the upper cervical muscles [26]. The activity of the afferent cervical spine is likely modulated by mechanoreceptors in the upper cervical spine [27]. The function of mechanoreceptors can be altered by direct trauma, muscular fatigue, degenerative changes, or direct effects of pain [28]. Therefore, upper cervical dysfunction can alter the orientation space and lead to dizziness and imbalances [7]. Borg-Stein et al. [27] also reported that CGD is associated with disturbances in the proprioceptors of the joints, muscles, and ligaments of the cervical spine. They noted that these dysfunctions provoke neck pain, myofascial pain, headache, dizziness, vertigo, imbalance, nausea, visual-motor sensitivity, ear fullness, tinnitus, or mood disturbances, such as anxiety or depression, and fatigue. However, in spite of strong connections between balance function and cervical receptors such as proprioceptors and joint receptors, CGD is excluded from the diagnosis because of the single diagnostic test and unique signs or symptoms [29,30]. In clinical practice, patients still complain of dizziness even though they have no cervical spine disease, vascular problem, or accident experience, as well as no vestibular lesion. Therefore, this paper suggests that structural and functional changes in the suboccipital muscles caused by abnormal head posture may be related to CGD.

## 2. Suboccipital Muscles and Dizziness

Understanding the function, structure, and role of the suboccipital muscles can be the first step to approach cervicogenic dizziness without trauma. The suboccipital muscles include the rectus capitis posterior major (RCPma), rectus capitis posterior minor (RCPmi), obliquus capitis superior (OCS), and obliquus capitis inferior (OCI) muscles. These were located in the deepest layer of the upper cervical spine [31,32,33]. The suboccipital muscles have higher muscle spindles per gram than other muscles in the upper cervical region. Muscle spindles are important for controlling movement and posture in mammals [31,34]. As such, high spindle densities (>50/g) are typically associated with small muscles that subserve fine motor tasks. These provide adequate proprioceptive information to control the position and movement of the head and to coordinate eye and head movements [31,35,36,37,38]. The suboccipital muscles have a high density of muscle spindles, which allow flexible movement and act as specific sensory receptors. However, Golgi tendon organ (GTO) is lacking [34]. The spindles densities in the suboccipital muscles are significantly higher than those in the splenius capitis, semispinalis capitis, and trapezius, which play similar roles in head and neck movement [34,39]. Among the suboccipital muscles, the oblique muscles (inferior oblique, 242; superior oblique, 190) are more than twice as many as the recti muscles (rectus capitis posterior major and minor, 98 each) [34]. These differences in spindle densities can be explained by the range of motion of the movement. The major movement of the atlantoaxial joint is rotation. The oblique muscles lengthen during almost the entire range of rotational movement at the atlantoaxial joint owing to their attachment location. Thus, they handle more proprioceptive input [34].

Human muscle spindles have distinct morphological characteristics that reflect the functional demands of muscles [31,40,41]. Liu et al. [31] observed a lack of bag2 fibers and wide variations in myosin heavy chain composition among intrafusal fibers. They speculated that spindles lacking bag2 fibers might have a relatively higher dynamic sensitivity, and they might reflect adaptation of the fusimotor system to the special task of controlling head posture and movements. They also mentioned that variations in myosin heavy chain seem to correlate with the variability in myosin ATPase activity noted along the length of individual nuclear chain fibers in the neck muscle spindles. The myosin heavy chain composition of intrafusal fibers is the best marker for the functional properties of human skeletal muscle fiber types because it correlates with contraction force and velocity [42,43,44]. Therefore, the distinct contents of these muscles might transmit complex proprioceptive inputs from the neck and head to the brain and act on mechanisms involved in head–eye coordination [34].

Muscles are generally classified as fast-twitch, intermediate, and slow-twitch types depending on their contractile and metabolic profiles [45]. The types of muscle fibers contribute to our understanding of their function and role [46]. The slow-twitch type is characterized by a slow contraction speed, oxidative metabolism, and fatigue resistance [45]. The muscles functioning as local stabilizers consist of slow-twitch type muscles with these features to provide the continuous activity required to maintain stability of the spine [47]. In addition, muscles that are short and closely related to the vertebral column are considered to be local stabilizing muscles, such as the lumbar multifidus [48]. Agten et al. [45] reported that the multifidus is a postural muscle that provides stability in the lumbar vertebral column because of its predominance of slow-twitch fibers. Yamauchi et al. [49] observed that RCPmi and OCS have at least two-fold greater cross-sectional areas of slow-twitch fibers than fast-twitch fibers in human cadavers, and that OCI has a significantly larger cross-sectional area of fast-twitch fibers than other suboccipital muscles. Therefore, they suggested that the RCPmi and OCS may provide more force than the RCPma and OCI and act as antigravity agonist muscles of the head. Richmond et al. [50] reported that slow-twitch fibers in OCI accounted for nearly all of the fibers in deep regions, but less than 10% of the fibers in the superficial fascicles. Cornwell et al. [46] also reported that the average number of slow-twitch fibers in elderly cadavers was 62.3%, and the range of suboccipital muscles was 58.8% (RCPmi) to 69.2% (OCI).

Although contraction of the suboccipital muscle is involved in head extension, unilateral head and neck rotation, and unilateral lateral flexion, these muscles are very small and attached close to the craniovertebral joint. Therefore, they show a mechanical disadvantage during movement in comparison with muscles making similar movements, such as the trapezius and splenius muscles [34]. The biomechanical function of short muscles located in the deep layers plays an important role in maintaining stability. They can be classified as joint stabilizers of the upper cervical spine or postural controllers [51,52,53]. Many researchers have reported that malformations, defects, and anomalies of the suboccipital muscles are associated with headache, neck pain, and CGD [49,54,55,56,57]. Therefore, understanding the function and role of the suboccipital muscles is an important first step in treating CGD. In summary, these muscles have the following characteristics: (1) they mostly consist of slow-twitch fibers involved in posture maintenance; (2) the attachment positions of these muscles are mechanically inefficient to induce movement and are of short sizes located in the deep layers; (3) they show high muscle spindle densities, special features of muscle spindles, and little GTO.

Therefore, the main functions of these muscles can be summarized as follows: (1) they may function as sensors transmitting sensory information with vestibular and ocular input and help monitor and recognize the upper cervical spine; (2) they act as stabilizers in the upper cervical spine rather than as operators of head and neck movement; and (3) they control the positions and precise movements of the head and help with eye–head movement coordination.

## 3. Abnormal Input of Suboccipital Muscles Caused by Forward Head Posture

Evidence of abnormal somatosensory input related to cervicogenic dizziness is as follows. The first is FHP. Continuous loading on the cervical spine, e.g., with extended smartphone or computer use, results in excessive thoracic kyphosis, degenerative disc disease, and FHP [58,59,60]. It is one of the commonly recognized types of poor head postures in the sagittal plane and involves changes in the surrounding soft tissues and cervical instability [21,61,62,63]. Various measurement tools are used for FHP: the craniovertebral angle, cervical inclination angle, and head tilt angle [61]. FHP enhances stress on the posterior cervical spine and soft tissues, affects the length–tension relationship in the cervical muscles, limits head and neck movement, and impairs cervical proprioception function [64]. With an increasing forward head posture, the anterior vertebral sagittal displacement is greater in the upper cervical spine than in the lower cervical spine. FHP appears to increase the anterior loading of articular facets [51]. The angle of the FHP and the weight transmitted to the cervical spine are proportional [65]. Hallgren et al. [51] studied the contraction level of suboccipital muscles during FHP. They reported that the mean EMG activity in the RCPm and RCPM muscles was approximately 10% to 18% of the maximum voluntary isometric contraction (MVIC) in the neutral head position, but it increased to approximately 34% to 42% of the MVIC in a FHP. An activity level of 30% of MVIC shows the endurance limit of force in sustained and static contraction of a muscle for approximately 1.0 to 2.5 min [51,66]. The upper endurance limit for 1 h is approximately 8% of the MVIC [66,67]. Björkstén and Jonsson [66] mentioned that the acceptable upper limit of force for a continuous (4–8 h) contraction is probably as low as a few percent of the MVIC. On the basis of these studies, it can be inferred that micro-damage is induced when excessive weight from the FHP accumulates in the surrounding structures. Such cumulative microtrauma can increase capsular ligament elongation by up to 70% of normal [68]. Damaged capsular ligaments generate abnormal ligament–muscle reflex responses [69]. The surrounding muscles exhibit abnormal contractions [51]. Capsular ligament laxity leads to excessive movement of the facet joint [9]. Changes in soft tissue continuously generate and accept inappropriate signals, eventually leading to cervical instability [70]. In general, clinical instability refers to the loss of motion stiffness caused by the force transmitted to a specific spinal segment, which causes a greater displacement than alignment of normal structures [9]. Cervical instability induces symptoms such as neck pain, stiffness, shoulder weakness, vertigo, dizziness, tinnitus, headache, and memory loss [9,71].

The second reason involves changes in the function and structure of the suboccipital muscles. FHP results in structural changes to the surrounding cervical muscles. The occipital extensors (longissimus capits, semispinalis capitis, sternocleidomastoid, upper trapezius, and most of the suboccipital muscles), except the splenius capitis, are shortened. The suboccipital muscles, except the obliquus capitis inferior, which acts in head rotation, undergo the greatest shortening [33,62,72]. Changes in muscle length caused by postural changes affect the binding between actin and myosin filaments, resulting in changes in muscle strength (i.e., force-generating capacity) and endurance (i.e., fatigue-resistant capacity) [73]. Uthaikhup et al. [74] also reported that structural changes in muscles may be associated with changes in fiber type, functional impairment (reduced strength and endurance), and altered postural and balance control. Functional or structural changes in the suboccipital muscles are associated with chronic headache, chronic neck pain, somatic dysfunction, and loss of standing balance [74,75,76]. Many studies have reported that patients with chronic neck pain or headache show marked atrophy of the rectus capitis posterior major and minor muscles, including fatty infiltration [74,75,76]. They suggested that the relationship between the structure (size and fatty infiltration) of the suboccipital muscles and symptoms, such as chronic headache or chronic neck pain, can be attributed to the following changes: (1) the lack of proprioceptive inhibition of nociceptors at the dorsal horn of the spinal cord would result in chronic pain and a loss of standing balance; (2) nociceptive inputs caused by active trigger points can lead to atrophy of the involved muscles; and (3) replacement of muscle tissue with fatty tissue may reduce the muscle spindle and GTO density. Fatty tissues may result in inaccurate feedback and the loss of important information related to muscle position and tension transmitted to the central nervous system. These changes in cervical muscles may have long-term consequences [77]. The activation changes of the deep neck muscles are compensated by increased activity of the superficial neck muscles, which show a delay in reaction velocity during postural disturbance and inhibitory signs of neck pain [78,79]. Prolonged overactivity of the superficial cervical muscles may have injurious effects on the properties of the muscle fiber membrane, resulting in greater muscle fatigability [77,80]. Fatigue of the superficial cervical muscles causes further weakening of the deep cervical muscles and promotes muscle fatigue. This vicious cycle continues with poor posture [81,82]. Localized muscle fatigue results in reduced muscle tension due to repetitive stimulation, which leads to a reduction in the ability to achieve optimal physical performance [83]. Localized muscle fatigue also reduces proprioception, which can lead to motor unit damage [84,85,86], and can decrease muscle conduction velocity, which results in decreased reaction time and accuracy, the ability to generate force, and muscle size (e.g., volume, cross-sectional area, length) [87,88].

The third reason is related to the myodural bridges of suboccipital muscles. Unlike the other deep muscles in the upper cervical spine, the suboccipital muscles have myodural bridges, which are connections between the dura mater and the suboccipital muscle fascia [89]. Contraction of the RCPma, RCPmi, and OCI muscles, which puts the myodural bridge under tension, transmit forces across it to place the dura under tension and stabilize the spinal cord [89,90,91]. Because the dura mater tends to fold inwardly on the spinal cord, myodural bridges play a role in resisting this movement [92]. Mechanical traction on the suboccipital tissues can cause movement of the dura mater at the cervicocranial junction [92,93]. Head posture deviating from the neutral head position affects the tension of the RCPma and RCPmi muscles, which in turn results in inability to maintain the tension of the myodural bridge [51]. Myodural biofeedback may help maintain the integrity of the subarachnoid space [90,94]. The central nucleus, which controls the suboccipital muscles, including the RCPma, RCPmi, and OCI muscles, may respond reflexively with feedback control of dural tension [95,96]. Therefore, myodural bridge issues are associated with cervicocephalic headaches, cervicocephalic pain syndromes, sensorimotor function, and postural control [89,93,97,98]. The suboccipital muscles might work as a pump via the myodural bridge to provide power for cerebrospinal fluid circulation [99,100]. The RCPmi myodural bridge prevents dural infolding when the head is extended or moved backward, inhibiting normal circulation of the cerebrospinal fluid. RCPma and OCI muscle myodural bridges also play similar roles to RCPmi [89,90,94,101]. Therefore, hypertrophy of the suboccipital muscles or failure to maintain constant tension in the myodural bridge can lead to increased dural tension, altered cerebral spinal fluid flow, and altered sensorimotor function, which can cause various clinical symptoms of tension [89,90,92,94,102].

The last is the activation of trigger points. Excessive release of acetylcholine by dysfunctional neuromuscular junction can form a taut band, which compresses the capillaries that supply nutrients to the muscle, which can cause muscle ischemia. This process does not provide enough energy to the working muscles, resulting in the release of inflammatory molecules, thereby activating nociceptive neurons [103,104]. Excessive muscle contraction produces metabolites (potassium, lactic acid, and arachidonic acid), and they directly sensitize proprioceptors. Incorrect proprioceptive input can increase activation of the muscles [105]. Therefore, a trigger point caused by excessive muscle contraction can lead to dysfunctional proprioception and nociceptor activation, and the vicious cycle can be linked to various clinical symptoms. FHP may activate trigger points, which are hyperirritable spots associated with a taut band of skeletal muscle, on the suboccipital muscles [106]. Activated trigger points on the suboccipital muscles can result in a referred pain pattern that spreads to one and/or both sides of the head above the occipital and temporal bones [106]. In addition, activated trigger points can transmit excessive nociceptive inputs to the central nervous system, which may cause maintenance or persistence of migraine [107,108]. Many researchers have reported that activation of trigger points in the suboccipital muscle might be attributable to activation of the caudate nucleus of the trigeminal nerve and the trigeminovascular system [107,109,110]. Repeated nociceptive activation originating from repetitive FHP might be delivered to the trigeminal nucleus, which would result in temporal and spatial summation of neuron signals. This can produce central sensitization and facilitate the trigeminovasular system. These processes will eventually lead to a decrease in the pain threshold, resulting in a gradual increase in headache [107,110,111]. Fernández-de-las-Peñas et al. [111] reported that the degree of FHP was positively correlated with headache duration, headache frequency, and the presence of suboccipital active trigger points.

Not all patients with abnormal head and neck posture complain of dizziness. However, if a patient with no problems in the vestibular system, cardiovascular system, and nervous system has an abnormal head and neck posture, structural and functional problems of the suboccipital muscles need to be investigated.

## 4. Conclusions

Incorrect posture alignment might cause CGD by the following process: FHP induces a change in the alignment and an excessive load on the upper cervical spine. These changes cause structural and functional changes in the surrounding muscles, especially the suboccipital muscles. In addition, unnecessary stimulation may persist because of the instability of the ligaments and facet joints. These alterations transmit abnormal proprioceptive inputs to the central nervous system, resulting in inconsistencies with vestibular and visual inputs. Mismatched information integration manifests as a variety of symptoms, including dizziness, pain, lightheadedness, and headache.

## Data Availability

Not applicable.

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
