# Peer review of "Suboccipital Muscles, Forward Head Posture, and Cervicogenic Dizziness"

_medicina, 2022, doi:10.3390/medicina58121791_

Round 1

Reviewer 1 Report

This is a well structured review work. The author thoroughly discusses available literature in the field, the issue is frequently discussed and such a review article provides an in-depth update on current scientific evidence in cervicogenic dizziness.

A minor grammar check would be in order. Other than that I found no serious issues, great work and congratulations. 

My decision is to accept as is.

Author Response

Thank you for your comments. 

I add editing certification.

Reviewer 2 Report

Dear Authors,

I think that the aim of the Manuscript to describe non-vestibular etiology of dizziness could be very interesting for scientists and clinicians all over the world. Minor English language corrections are required.

Good luck! :)

Author Response

thank you for your comments.

I add editing certification.

Reviewer 3 Report

Review of «Suboccipital muscles, Forward head posture, and Cervicogenic dizziness»

Summary

This article is a narrative review. It reports (in a non-systematic fashion) literature on proposed pathophysiological mechanism behind cervicogenic dizziness (CGD). The report contains considerations regarding the potential link between upper cervical spine and dizziness, suboccipital muscles and dizziness, forward head posture (FHP) and suboccipital muscles. It concludes by proposing a new factor in CGD: FHP.

The article is fairly articulated around the logical inducive idea that FHP contributes to afferent proprioceptive input changes that could result in a sensory mismatch which in turn might contribute to dizziness. Generally, the article is fairly relevant for the field and represents the selected data from the selected literature. Although, it is sometimes presented in a relatively poor manner, it does fit the broad scope of the Medicina journal.

General concept comments

In general article needs more specific and clearer information. It also needs more support in terms of reference for claims made and a more explicit articulated inducive process. The authors address an important question about non-specific dizziness and the complexities of CGD. It would benefit from more nuanced way to present this working hypothesis.

More specific and detailed description of population is needed in order for the question to be better defined: CGD population should be presented in terms of diagnosis criteria (clear exclusion diagnosis process should be explained) (Line 31-34). Article should also consider alternative hypotheses and explanation for the symptoms and discuss why the FHP is the best hypothesis (vascular origin of CGD is not discussed for example). Clear diagnosis criteria for FHP should be mentioned (what tests are clinicians supposed to do? Is it only a clinical impression?) Specifications on how the FHP factor hypothesis provides an advancement of the current knowledge should be clearly defined and appropriate. How could FHP be investigated in relation to CGD in order to advance knowledge. Most importantly, the question of why does FHP results sometimes in dizziness, and sometimes patient with FHP don’t present dizziness should be hypothesized.

Rating the Manuscript

This is a narrative review and does not have any reproducibility or transparency.  Some methodological background for the selection of articles or information about how did this article come to be would enhance the quality of the article: Why wasn’t a systematic approach considered or if it was, why wasn’t it done should be discussed.

The abstract (line 8 to 17) should contain a summary of the article or at least state the different part of the structure of this narrative review. The purpose of the article mentioned in the abstract is to «diagnose and treat patients with non-specific dizziness» (Line 16-17). This is misleading because this article is merely a narrative review and clinical reasoning and care trajectory should rely on higher evidence level papers like clinical guidelines, meta-analysis, well designed randomized control trials or strongly powered longitudinal observational designs.  The goal of the article could be: to propose an explanatory framework for dizziness of cervical origin in the absence of trauma, degenerative disease, vascular lesion of the neck and chronic cervical pain having also eliminated other probable causes of dizziness.

The review should be more comprehensive even for a narrative review: the population should be better defined. For instance: CGD is a subset of non-specific dizziness that remains elusive because its diagnosis relies on the exclusion of other potential causes. This criterion is not mentioned in the article. That impacts the external validity of the article because the population is not well defined. How does one suggest a new risk factor for a clinical syndrome (CGD) that has not been well defined for the readers? (Lines 27 to 37).

Although the articles cited are pertinent, they are often very few references to support the claims of the article (most ideas are supported by only one reference and some by none). Line 21-22, Line 46-47, Line 66-67, Line 133-138, Line 171-174, Line 201 should have at least one reference. Self-citations should be limited to 1 or 2 instances in the article, other article can support the claims. Limiting self-citation preserves an impression of scientific neutrality. In line 133, GTO is not defined anywhere in the article.

Line 43, «Postural sway» is more adequate to «postural dizziness» as it makes the sentence make sense.

Line 50-52 is incorrectly supported by the references : vestibula doesn’t inform on orientation of the head with respect to the rest of the body. Vestibula informs in relation to acceleration vectors including gravity (i.e. acceleration of head in space without regards to the position of the body). Visual clues can only inform on orientation of the head with respect to the rest of the body if the body is in the visual field. Generally speaking, only proprioceptive cues can inform about orientation of the head with respect to the rest of the body.

Line 208-226: It is not clear how myodural bridge is related to dizziness.

Line 227-242: It is not clear how trigger points and pain is related to proprioceptive input changes and potentially dizziness. Consideration for migraine and headache is interesting, but not apparently pertinent. How does nociceptive input change proprioceptive input should be discussed and included in Figure 1.

Line 257 to 277, presents an analogical model of a video camera to illustrate how visual, vestibular and proprioceptive afferents are integrated with the cerebrum and cerebellum. Although an explanatory model is interesting in a pedagogical way, the one proposed here is not pertinent specifically to this subject and it lacks precision. These lines should be omitted.

In Figure 1, «Other» should be defined and pain should also affect proprioceptive input.

Author Response

Thank you for your comments.

I added and deleted some sentences based on your comments. The corrected part is marked in red.

Round 2

Reviewer 3 Report

The Authors responded to our concern.

it  needs a revision by an english speaking native.